# Endothelial Cell Metabolism in Vascular Functions

**DOI:** 10.3390/cancers14081929

**Published:** 2022-04-11

**Authors:** Antonio Filippini, Luca Tamagnone, Alessio D’Alessio

**Affiliations:** 1Sezione di Istologia ed Embriologia Medica, Dipartimento di Scienze Anatomiche, Istologiche, Medico Legali e dell’Apparato Locomotore, Sapienza Università di Roma, 00161 Roma, Italy; antonio.filippini@uniroma1.it; 2Sezione di Istologia ed Embriologia, Dipartimento di Scienze della Vita e Sanità Pubblica, Università Cattolica del Sacro Cuore, 00168 Roma, Italy; luca.tamagnone@unicatt.it; 3Fondazione Policlinico Universitario “Agostino Gemelli”, Istituto di Ricovero e Cura a Carattere Scientifico (IRCCS), 00168 Roma, Italy

**Keywords:** angiogenesis, caveolin, COVID-19, endothelial cells, endothelial dysfunction, metabolism, vascular disease, vasculogenesis

## Abstract

**Simple Summary:**

Recent findings in the field of vascular biology are nourishing the idea that targeting the endothelial cell metabolism may be an alternative strategy to antiangiogenic therapy, as well as a novel therapeutic approach for cardiovascular disease. Deepening the molecular mechanisms regulating how ECs re-adapt their metabolic status in response to the changeable conditions of the tissue microenvironment may be beneficial to develop novel innovative treatments to counteract the aberrant growth of vasculature.

**Abstract:**

The endothelium is the innermost layer of all blood and lymphatic vessels composed of a monolayer of specialized endothelial cells (ECs). It is regarded as a dynamic and multifunctional endocrine organ that takes part in essential processes, such as the control of blood fluidity, the modulation of vascular tone, the regulation of immune response and leukocyte trafficking into perivascular tissues, and angiogenesis. The inability of ECs to perform their normal biological functions, known as endothelial dysfunction, is multi-factorial; for instance, it implicates the failure of ECs to support the normal antithrombotic and anti-inflammatory status, resulting in the onset of unfavorable cardiovascular conditions such as atherosclerosis, coronary artery disease, hypertension, heart problems, and other vascular pathologies. Notably, it is emerging that the ability of ECs to adapt their metabolic status to persistent changes of the tissue microenvironment could be vital for the maintenance of vascular functions and to prevent adverse vascular events. The main purpose of the present article is to shed light on the unique metabolic plasticity of ECs as a prospective therapeutic target; this may lead to the development of novel strategies for cardiovascular diseases and cancer.

## 1. Introduction

From a histological viewpoint, the term endothelium refers to the simple squamous epithelial tissue composed of specialized ECs that lines the luminal surface of all blood and lymphatic vessels. This definition has conferred for a long time the endothelium the modest role of a simple as well inert physical epithelial barrier interposed between the blood and the surrounding tissues. However, the growing interest in the field of vascular biology that we have witnessed over the years has contributed to promote the endothelium to a genuine multifunctional dynamic organ that participates to countless physiologic processes, including the movement of nutrients between the blood and the surrounding tissues, the extravasation of immune cells and inflammation [1,2], the control of hemostasis and coagulation [3,4], the regulation of vascular tone [5] and vascular permeability [6,7], and the guidance of the wound healing process and the vascular remodeling during angiogenesis [8,9]. Endothelial dysfunction is a general term that describes a disturbance in the normal functions of ECs, leading to a multiplicity of pathological conditions (Figure 1).

ECs cover hundreds of square meters of blood vessels, establishing an extensive vascular bed organized into a hierarchical branched framework of arteries, arterioles, capillaries, venules, and veins perfusing almost every tissue throughout the body [10]. Across the human body, only the cartilages, lining epithelial tissues, and lens are devoid of a genuine vascular network, which makes these tissues highly dependent on the adjacent connective tissues for their nutritional, healing, and regenerative needs. Along the vascular tree, the set of larger vessels, such as arteries and veins, are typically referred to as the macrivasculature, which participates in the movement of blood toward or away from organs, while arterioles, capillaries, and venules form the microvasculature that is responsible for the metabolic exchanges between blood and the neighboring tissues. Although ECs share several properties, they also display a fair degree of heterogeneity both at the structural and functional level and exhibit different gene expression profile depending on the anatomical site [11,12,13,14,15]. Capillaries are the smallest and most numerous of the blood vessels, consisting of a single layer of ECs with occasional surrounding mesenchymal cells endowed with contractile functions known as pericytes (also known as Rouget or mural cells) [16,17,18]. Interestingly, recent studies have highlighted the role of pericytes in the onset of vasculopathies and pro-thrombotic conditions in coronavirus disease 2019 (COVID-19)-positive patients [19], endorsing the role of these perivascular cells as a potential therapeutic target in vascular diseases and neurological disorders [20,21,22,23]. In addition, three types of capillaries, defined as continuous, fenestrated, and discontinuous, exist, characterized by variably sized gaps between ECs. In continuous capillaries, ECs lack transcellular perforations, are joined by tight impermeable junctions, and produce a continuous basal membrane. Continuous capillaries are common in tissues with high metabolic requirements such as nerve (where they form the blood–brain barrier) and muscle tissue [16]. In fenestrated capillaries, ECs display 80 to 100 nm pores (fenestrae) spanned with thin diaphragms that grant these blood vessels a higher permeability than continuous capillaries. They are typically found in tissues involved in absorption such as the intestine and kidney and in some endocrine glands [24]. Discontinuous, sometimes referred to as sinusoidal capillaries show open spaces between ECs, display either an irregular or absent basement membrane, and are highly permeable, allowing the movement of serum proteins and even whole cells between blood and tissues. Discontinuous, capillaries are typically found in the liver, spleen, and bone marrow [16,25]. In addition, modified postcapillary venules, typical of lymph nodes, known as high endothelial venules (HEVs), show a unique cuboidal epithelium rather than the conventional squamous one, and represent a distinctive example of morphological heterogeneity amongst ECs [26]. Capillaries are specialized in the exchange of material between blood and the surrounding tissues, whereas the post-capillary venules are actively involved in the response to inflammatory stimuli, guaranteeing the appropriate trafficking of circulating leukocytes to the surrounding tissues. Arteries and veins are composed of three tissue layers (i.e., tunicae) containing a precise arrangement of epithelial, connective, and smooth muscle tissue. The functional diversity of ECs is also demonstrated by the different response activated in situ by arteries and veins to inflammatory stimuli [27], as well as by the release of tissue-specific paracrine angiocrine factors contributing to homeostatic functions, organ regeneration, and pathological conditions [28,29,30].

## 2. The Origin of the Endothelial Cells

The blood circulatory system begins to form approximately at the end of the third week of development when the diffusion of nutrients and oxygen through the amnion, extraembryonic coelom, and yolk sac no longer meets the nutritional demands of the developing embryo. The earliest vascular structures in the embryo arise in the extraembryonic splanchnic mesoderm of the yolk sac, the connecting stalk, chorion, and later in the intraembryonic mesoderm as undifferentiated mesodermal cells aggregates known as blood islands [31,32,33]. Once formed, neighboring blood islands merge to form a primitive capillary plexus that further develops into a mature circulatory network [34] through vasculogenesis, i.e., the de novo formation of blood vessels by differentiation of individual intra- and extra-mesodermal endothelial progenitor cell (EPC) precursors or angioblasts [35,36,37,38]. Cells within the blood islands occur with a definite phenotypic signature, i.e., peripheral EPCs or angioblasts differentiate into ECs, whereas cells inside the islands designated hematopoietic stem cells (HSCs) become the primitive blood cells. Vasculogenesis is the central mechanism by which the primordia of the first embryonic vessels such as the dorsal aorta and the endocardium form. New capillary branches can now develop from the pre-existing primitive vascular plexus through sprouting angiogenesis and vascular remodeling, expanding the initial vessel network within the embryo [39,40]. After birth, angiogenesis is physiologically restricted to the uterine cycle, pregnancy, wound healing, and it is induced by constant workout of the skeletal muscle (or it can regress in inactive individuals) as well as in adipose tissue because of body weight gain (or reduced when it decreases again), and it is activated during tissue injury, arterial occlusion, inflammation, and cancer [41,42]. Typically, adult neo angiogenesis initiates with the proteolytic degradation of the basement membrane and pericyte detachment from the existing blood vessels, resulting in EC proliferation and migration towards the angiogenic stimulus, such as vascular endothelial growth factor (VEGF). The laying down of fresh extracellular matrix and the recruitment of pericytes results in the maturation of the new sprout into a mature vessel or a new capillary bed when it merges to other adjacent nascent sprouts. During the final step of vasculogenesis, termed tubulogenesis or lumen formation [43], angioblast-derived ECs acquire their typical flat morphology and establish their definitive junctional contacts, resulting in the initiation of blood flow [44] (Figure 2). However, we must not forget that angiogenesis and vasculogenesis are crucial mechanisms that cooperate in neovascularization and contribute to vascular diseases in adulthood. Indeed, the ability of circulating adult bone marrow-derived immature EPCs [45] to drive postnatal vasculogenesis and re-endothelialization has encouraged the development of EPC-based therapies to treat vascular diseases in adults [46,47,48,49]. In addition, the biological activity of EPCs and their differentiation to functional ECs may be related to the specific metabolic status they acquire during recruitment in response to specific stimuli. After all, both angiogenesis and the maintenance of the stemness feature are regulated by hypoxia, and the choice of stem cells to rely on glycolysis is pivotal to avoid the dangerous effect caused by the reactive oxygen species generated during mitochondrial respiration. To date, about 400 studies have been registered under the term “EPC” in the clinical trial registry (www.clinicaltrials.gov, accessed on 26 February 2022), focusing on different pathological conditions including ischemia, stroke, cancer, pulmonary disease, and treatment-resistant depression.

## 3. Angiogenesis in Cancer

In contrast with what occurs during intrauterine life, in adulthood, ECs rarely divide or sprout. Therefore, blood vessels remain largely quiescent, but preserve a remarkable ability to respond to physiological and pathological cues. This stability of adult vasculature comes in part from the close association of mural cells with the abluminal surface of ECs. Adult angiogenesis remains restricted to few physiological and highly regulated events, including the uterine cycle, the regeneration of damaged tissues during the wound repair and healing process, and throughout the development of the placenta in pregnancy. As a result, angiogenesis requires a dynamic relationship between pro- and anti-angiogenic factors, and any mechanism that disrupts this balance can lead to disease. In brief, certain pathological conditions such as diabetic eye disease, rheumatoid arthritis, and cancer are featured by excessive angiogenesis and vascular growth. Conversely, ischemic tissue injury, cardiac failure, coronary heart disease, and delayed wound healing result from deficient angiogenesis. In 1971, Folkman hypothesized that tumor growth is angiogenesis-dependent, predicting that tumor masses would not be able to grow beyond 1–2 mm^3^ and remained dormant without the recruitment of new blood vessels. The results of these studies were consistent with the ability of tumor cells to secrete angiogenic factors, envisaging the use of antiangiogenic molecules as an anti-cancer strategy [50]. Following the discovery that many tumors secrete VEGF and express a high level of VEGF mRNA, it is not unexpected that the VEGF pathway has been considered one of the most attractive targets for the development of anti-angiogenic drugs. As a result, VEGF inhibitors and anti-VEGF receptor antibodies have been developed clinically as anti-cancer agents (Table 1). Although the scientific rationale underlying the use of angiogenesis inhibitors appeared to be promising in prolonging progression-free survival, the therapeutic benefits for the patients have not been completely successful. Toxicity and acquired tumor resistance through the activation of alternative angiogenic pathways that sustain tumor vascularization and growth are major disadvantages in anti-angiogenesis therapies, particularly in cancers that rely on pathways other than VEGF [51]. Indeed, the recruitment of angiogenic factors such as angiopoietins, epidermal growth factor (EGF), placental growth factor (PGF), and hepatocyte growth factor (HGF) capable of triggering alternative pathways leading to blood vessel formation can significantly contribute to the failure of anti-angiogenic therapy [52]. Likewise, stromal cells present within the tumor microenvironment such as cancer-associated fibroblasts (CAFs), bone-marrow-derived cells, and EPCs endowed with pro-angiogenic capability can further enhance tumor resistance to common anti-angiogenic therapies [53]. Therefore, it is urgent to develop innovative strategies able to bypass the weakness of current therapies in the treatment of cancer. To this regard, EC metabolism has recently emerged as a novel mechanism, in addition to the growth factor-dependent one, contributing to blood vessel formation. On the basis of the different metabolic profile exhibited by EC subtypes during vessel sprouting [54] as well as by tumor cells [55], targeting of specific metabolic pathways may be of help to counteract the failure of current anti-angiogenic therapies.

## 4. Overview of Cellular Metabolism

Cells respond to a variety of extracellular cues by activating specific intracellular signal transduction pathways as well as mid- to long-term gene expression profiles. At the same time, cells need to rapidly switch between different metabolic pathways to adapt to the variable surrounding cell microenvironment. The term metabolism refers to the set of processes in living cells that generate or consume energy, encompassing two major pathways referred to as catabolism and anabolism. Catabolism refers to the degradative route of metabolism, through which larger molecules (proteins, polysaccharides, fats, and nucleic acids) are broken down into smaller units, resulting in the release of chemical energy stored in the form of adenosine triphosphate (ATP) [56], and a small portion that is lost as heat and waste products. Energy produced during catabolism is used to fuel anabolic paths aimed at building complex macromolecules from simple nutrients, such as amino acids, monomers of carbohydrates, nucleotides, and fatty acids [57,58,59] (Figure 3). Glucose is the typical fuel substrate of cell metabolism, although other macromolecules including amino acids, fatty acids, and occasionally nucleic acids can be used to obtain chemical energy through glycolysis, the citric acid (Krebs) cycle, and oxidative phosphorylation [60]. In terms of energy supply, the oxidation of glucose in eukaryotic cells produces a theoretical yield of 38 ATP molecules, both in the presence of oxygen (aerobic catabolism) and in its absence (anaerobic catabolism), although the latter is much less efficient. Glycolysis is the first stage of glucose breakdown that occurs anaerobically in the cell cytoplasm and leads to the production of two molecules of pyruvic acid and two molecules of ATP. The remainder 36 ATP molecules are retrieved in the presence of adequate oxygen level when the two molecules of pyruvic acid (following their conversion to acetyl-coenzyme A, or acetyl-CoA) are further processed during the Krebs’ cycle and the oxidative phosphorylation in the mitochondria. In the absence of oxygen, pyruvic acid is transformed to lactic acid (which is pumped out of the cell) and only if oxygen levels are restored can it be converted to pyruvic acid to fuel back the Krebs’ cycle. In addition to glucose, amino acids are potential metabolic fuel during starvation that can be either converted to glucose to foster glycolysis or transformed to intermediate compounds such as acetyl coenzyme A to sustain the Krebs’ cycle and oxidative phosphorylation. Similarly, in the absence or reduced levels of glucose, cells can also rely on fat stores, converting fatty acids to acetyl coenzyme A, and glycerol to glyceraldehyde 3-phosphate, fueling the Krebs’ cycle and glycolysis, respectively. It should be noted that the capability of eliminating the unwanted and potential detrimental substances such as carbon dioxide (CO_2_), water, urea, uric acid, and ammonia, arising from metabolic activities, is as well crucial for cells and requires an accurate waste disposal pathway [60,61]. Undoubtedly, a precise balance between anabolism, catabolism, and waste removal is essential to guarantee cells and tissues homeostasis. For a more exhaustive overview of reactions, metabolites, and metabolic pathways please refer to the following link, http://www.genome.jp/kegg/pathway.html (acceded on 26 February 2022). It is worth mentioning that the development of a complex intramembranous system in eukaryotic cells provide an additional level of regulation of the different metabolic processes compared to that of prokaryotes [62,63]. In eukaryotic cells, most biosynthetic processes are typically restricted into defined organelles, such as mitochondria, while catabolic pathways remain predominantly cytoplasmic. By contrast, in prokaryotes that lack specialized organelles, the control of metabolism remained inevitably less refined. We can predict that the inability of cells to perform their normal biochemical reactions show a different extent of severity, depending on the affected pathway, eventually leading to the onset of pathological conditions. Therefore, the in-depth understanding of the mechanisms involved in the regulation of metabolic processes is crucial to identify novel molecular targets aimed at hindering metabolic syndrome.

## 5. The Basis of Endothelial Cell Activation

The first mention to the term endothelial activation dates to the late sixties, when Willms-Kretschmer described the ultrastructural changes occurring in cultured ECs in response to T-cell-induced delayed hypersensitivity reactions [64]. Additional evidence generated during the following two decades contributed to the identification of cytokines responsible for endothelial changes observed in vitro [65,66,67]. In the late 1980s, Pober defined the term endothelial activation as “*quantitative changes in the level of expression of specific gene products (i.e., proteins) that, in turn, endow endothelial cells with new capacities that cumulatively allow endothelial cells to perform new functions*” [68]. The typical signs of endothelial activation include cytokine production [69]; the increased expression of surface E-selectin, ICAM-1, and VCAM-1 [70]; elevated vascular permeability; and the gain of a prothrombotic phenotype [71,72,73]. The increasing number of studies on vascular functions contributed to the identification of several EC activation biomarkers, often converging into similar intracellular mechanisms and transducers, of whom the nuclear transcription factor κB (NFκB) is undoubtedly the most recruited one, particularly during inflammation [2,74,75]. Likewise, adult ECs can be activated to acquire an angiogenic phenotype, resulting in the initiation of sprouting angiogenesis, in oxygen-deprived tissues. One of the major mediators of EC activation during angiogenesis is VEGF that triggers cell responses by recruiting tyrosine kinase receptors VEGFR1/Flt-1, VEGFR2/KDR/Flk-1, and VEGFR3/Flt-4. VEGFR2 is the most potent mediator of changes that occur during VEGF-induced tip cells selection, while VEGFR-1 acts as a decoy receptor for VEGF [76]. VEGFR3 is mainly involved in lymphatic endothelial migration and proliferation [77,78], as well as in the regulation of VEGFR2 signaling and vascular permeability [79]. The key role of VEGF-mediated signaling in blood vessel formation was demonstrated by the discovery that heterozygous knock-out (VEGF^+/−^) mice developed significant vascular defects along with impaired ECs differentiation and abnormalities in angiogenesis and tubulogenesis, leading to early death during embryogenesis [80,81]. During the first stage of sprouting angiogenesis, VEGF promotes a rivalry between ECs to become leader tip cells or be intended to remain trailing stalk cells of the developing sprout. Once established, non-proliferating and high migratory tip cells protrude out the vessel wall by projecting extensive filopodial processes, while staying anchored to the originating blood vessel. The early phase of the angiogenic process requires the dissolution of the specialized type IV collagen and laminin rich connective tissue endothelial basement membrane by specific membrane-type matrix metalloproteinases [82], allowing ECs to form the first sprout [83,84,85], while the proliferation of stalk cells contributes to its elongation [86]. Notably, the tip or stalk phenotype is attributed transiently to competing ECs and depends both on the ability of tip cells to sense VEGF and the activation of the Notch signaling [87]. In mammals, the Notch pathway is involved in a variety of processes and consists of five transmembrane ligands (Jagged 1, 2; Delta-like ligand 1, 3, and 4) that bind to four different transmembrane receptors (Notch 1, 2, 3, and 4) [88]. The leading tip cells that guide the nascent sprout respond to VEGF by inducing the expression of Notch ligand Delta-like 4 (Dll4), whom binding to Notch receptor on the neighboring stalk cells inhibits VEGFR2 pathway and represses any attempt for their part to assume the tip cell phenotype [89,90,91]. Thanks to its ability to determine the tip cell phenotype, Notch signaling is a promising target for dampening aberrant angiogenesis [92,93,94]. Although VEGF is an undisputed crucial player of EC activation for both physiological and pathological angiogenesis, other factors have been found to contribute to a well-functional vasculature. Among these, semaphorin- and plexin-mediated signaling, previously identified as key mechanisms of axon guidance, are known to play significant roles in vascular shaping [95,96]. This is not surprising, since the vascular and neuronal system share a similar pattern of development [97]. In addition, it must be noted that filopodia that emanate from endothelial tip cells and the axonal tip that leads to its extension are morphologically similar and share the expression of some receptors [98,99,100]; this is noteworthy for guidance signals of two such systems that perform distinct roles within the body. Membrane-bound and secreted semaphorins signal through plexin and neuropilin receptors exerting either pro- or anti-angiogenic effects, making these pathways potential therapeutic targets for pathological angiogenesis [101]. In addition, other molecules and signaling pathways, such as slits and roundabout (Robo) receptors, netrin/deleted in colorectal cancer (DCC) and netrin/homolog UNC5 (UNC5) pathways, have been reported to play a role during both vascular and nerve tissue development [102,103]. Intracellular calcium increase plays a pivotal role in ECs, regulating a multiplicity of vascular functions [104]. Of note, the contribution of specific intracellular calcium mobilization from acidic stores through the acid adenine dinucleotide phosphate (NAADP)-dependent mechanism has been linked to VEGF-induced angiogenesis, adding a level of complexity to the already intricated regulation of VEGF signaling [105]. Undoubtedly, the tight balance between pro- and anti-angiogenic factors, along with the components of the microenvironment, are crucial to control neo-angiogenesis and vascular patterning under normal and pathological conditions. Despite the role of cellular metabolism recently appearing as a new regulatory player of endothelial functions [106], it deserves a deepened investigation to frame its precise contribution in vascular conditions.

## 6. Metabolic State of the Endothelial Cell in the Adult Vasculature

Due to the strategic positioning of the endothelium, ECs have earned a front-row seat to interact with blood, the largest source of oxygen and nutrients in the body [107]. From a metabolic viewpoint, ECs are largely glycolytic [54,106,108,109], that is, they do not take advantage of this ready-to-use bulk of oxygen to fuel the Krebs’ cycle and oxidative phosphorylation (OXPHOS). In fact, ECs retrieve more than 85% of their energetic needs from glycolysis, while only a small amount of pyruvate generated during this process is used to fuel OXPHOS, whose rate has been estimated to be about 200-fold lower than glycolysis in human umbilical vein ECs (HUVECs) [54,106]. In this regard, ECs behave like cancer cells, which prefer to convert glucose to lactate, even in the presence of oxygen, generating a smaller amount of ATP compared to OXPHOS. This mechanism is known as aerobic glycolysis or the Warburg effect, in recognition of Otto Warburg, who first observed in the early 20th century the rapid fermentation of glucose by tumors in the presence of abundant oxygen [110]. It is important to note that although under normal conditions ECs remain quiescent for extended periods, they are still able to rapidly adopt a more active phenotype in response to specific proangiogenic and proliferative prompts [111]. The choice of relying on glycolysis allows ECs to deliver more oxygen to the surrounding tissues as well as to minimize the amount of harmful reactive oxygen species (ROS) compared to OXPHOS. In addition, while mitochondrial OXPHOS is more efficient in terms of yield of ATP molecules, glycolysis is faster in terms of ATP synthesis [112,113]; this is crucial to ECs, especially during the early phases of angiogenesis when two distinct EC metabolic phenotypes, namely, tip and non-tip cells, differentiate within the new sprout. Notably, although these two EC subtypes can sustain both glycolysis and OXPHOS, CD34^+^ tip cells appear decisively less glycolytic and show a reduction in glucose uptake than CD34^−^ non-tip cells that, by contrast, mostly rely on mitochondrial respiration for proliferation [114]. However, while most studies in this field have focused on the molecular mechanisms that regulate the switch of ECs to an active phenotype, how their quiescent condition is regulated or reestablished after cells have fulfilled their task has been poorly investigated. Studies that emerged in the last decade have contributed to shedding light on this mechanism. In HUVECs, the suppression of nuclear-factor (NF)-κB activation through an Erg-dependent mechanism, resulting in the impairment of proinflammatory genes, has been shown to contribute to the maintenance of ECs quiescence [115]. More recently, the contribution of FOXO1 [116] and Notch signaling [117] were discovered as important regulators of EC proliferation and quiescence. These data agree with earlier findings demonstrating the involvement of FOXO1 in reducing the oxidative metabolism of HUVECs in vitro [118]. In addition, the expression of several molecular regulators of glycolysis such as phosphofructokinase-1 (PFK1), 6-phosphofructo-2-kinase/fructose-2,6-bisphosphatase 3 (PFKFB3), and hexokinase 2 (HK2) has been directly involved in the metabolic switch of ECs. The contribution of Krüppel-like factor 2 (KLF2) in the regulation of glycolytic enzymes and its contribution to sustain the resting state of ECs has also been reported [119]. Considering the role of endothelial dysfunction and the emerging role of cell metabolism in regulating EC normalcy [120,121], it follows that deepening at the molecular level the relationship between these two mechanisms will be decisive in the field of vascular biology and cardiovascular disease.

## 7. Endothelial Cell Metabolism, Angiogenesis and Inflammation

To our knowledge, the first historical reference to the term metabolism dates to the studies of circulatory physiology by Ibn al-Nafis in the 13th century [122]. However, cell metabolism was deeply studied across the 19th and 20th centuries when most biochemical reactions and enzymes involved in metabolic pathways were identified [123,124,125]. Nevertheless, it is only recently that we have started to focus on the distinct endothelial metabolic signatures of EC subtypes. As mentioned above, the unique metabolic plasticity displayed by ECs not only is crucial to fulfilling the set of physiological processes of tissue remodeling but is accountable for the abnormal behavior of ECs during pathological angiogenesis. Quiescent (non-cycling) ECs, particularly those of capillary and post-capillary venules, possess a well-equipped molecular toolbox to perceive a variety of angiogenesis-inducing signals. Hence, keeping the precise balance of inducers and inhibitors of angiogenesis is crucial to avoid aberrant, i.e., excessive or insufficient, neovascularization that could lead to the onset of pathological conditions, including skin diseases [126], age-related macular degeneration [127,128], diabetes [129,130], cardiovascular disease [131,132,133,134], and cancer [40,135,136,137,138]. In addition to the well-known contribution of pro- and anti-angiogenic factors to the angiogenic process, it is emerging the idea that ECs can rapidly switch between specific metabolic pathways in response to changes in the extracellular environment [121]. In this regard, PFKFB3, a major regulator of the glycolytic process, is crucial in inducing the migratory phenotype of tip cells [54]. By contrast, the stalk cell phenotype appears to rely on fatty acid oxidation [139]. Unstimulated ECs display a typical phalanx (a word used to describe the Greek soldiers’ military formation) phenotype of firmly interconnected cells that is pivotal to keeping the endothelium in a quiescent condition, ensuring the proper oxygen delivery to the surrounding tissues, and it has been suggested to impair the formation of tumor metastasis [140].

## 8. Endothelial Cell Metabolism and Viral Infection

Viruses are the smallest infectious agents that, unlike animal cells, lack the biological machinery necessary to produce metabolic energy and are not self-sufficient in replication. This functional limitation obliges viruses to infect a suitable host cell, such as animal or plant cells, protists, fungi, and bacteria (or bacteriophages), coercing it to synthesize virus-specific macromolecules needed for the virus to be replicated. The investigation of virus-infected cell metabolome has revealed the capability of these pathogens to modify certain metabolic pathways of the host cell, including glycolysis, glutaminolysis, and fatty acid synthesis [141,142]. For instance, upon infection, human cytomegalovirus [143], human adenovirus [144], and the hepatitis C virus [145] increase glucose uptake and glycolysis, resulting in high lactate production of the host cell [146]. While the induction of the Warburg effect in tumors is likely aimed at protecting cancer cells from the dangerous reactive oxygen species (ROS) that are largely produced by mitochondrial respiration, it results in being pivotal for the replication of the viral genome or for virion envelopment within the host cell [146]. Notably, because of the strategic positioning of endothelia, one can predict that the capability of a given virus to infect ECs is crucial for its propagation to other organs. Indeed, it has been demonstrated that many viruses that target ECs can impact vascular functions [147,148]. The talent of Zika virus to increase the cellular glycolytic rate by inhibiting AMPK signaling to sustain its own replication has been recently shown in human retinal vascular ECs [149]. Moreover, members of Herpesviridae, a widely distributed family of DNA viruses that establish a latent and persistent infection in ECs, induce glycolysis and glucose uptake to guarantee cell survival of latently infected cells [150,151]. Several types of viruses, including henipaviruses, hantavirus, avian influenza virus, and the recently discovered single-stranded RNA virus SARS-CoV-2 can infect human ECs [148,152,153]. To this regard, the emerging role of ECs in COVID-19 is of great interest, considering the ability of SARS-CoV-2 to cause a general breakdown of vascular functions, especially in patients with pre-existing endothelial-related dysfunction such as diabetes, hypertension, and other cardiovascular disorders [154,155,156] that frequently exacerbate COVID-19 pathology. On the basis of studies conducted in COVID-19 patients, as well as from the analysis of post-mortem histological samples, it appeared a significative correlation between SARS-CoV-2 infection and the development of thrombotic conditions, resulting from a cooperation of the immune and vascular systems in the effort to recognize and neutralize the pathogen, a general mechanism referred to as immunothrombosis [157]. In this respect, we may wonder whether targeting host metabolic pathways may be useful to control the outcome of specific virus infection, both in ECs and other cell types. To address the issue, we should investigate in depth the relationship between virus infection and host cell metabolism, with the aim to identify novel metabolic targets that may be of help to prevent detrimental virus effects on the host, especially when available vaccines are not fully effective at protecting against disease, as in the case of the current COVID-19 pandemic. Recently, targeting the endo-lysosomal two-pore channel 2 with naringenin has been disclosed as possible antiviral therapy in in vitro SARS-CoV-2 infection [158].

## 9. Involvement of Plasma Membrane Microdomains to Endothelial Cell Metabolism

The plasma membrane (PM) of some cell types is provided with characteristic small microdomains particularly enriched in cholesterol and sphingolipids called lipid rafts (LRs) [159,160]. Although the existence of LRs in vivo is debated, it is generally accepted that their unique lipid composition gives rise to a unique signaling platform capable of regulating, i.e., compartmentalizing, certain signaling molecules [161,162]. Notably, in some cell types, LRs can further organize into 50–100 nm clearly detectable, non-coated, small, flask-shaped invaginations termed caveolae, characterized by a lipid composition like that of LRs. Since the first discovery of caveolae in the early 1950s [163,164], the lack of specific molecular markers largely restricted the investigation of these organelles to electron microscope observations, leaving open the debate on the biological role of the caveolar network for a long time. However, following the identification in the 1990s of the caveolin proteins [165,166,167,168] known as the first molecular markers as well as the main scaffolding proteins of caveolae, and the identification of cavins, as crucial proteins contributing to caveolae biogenesis [169,170], the interest towards these fascinating organelles as prospective regulators of cell functions has gradually increased. Caveolae are particularly abundant in ECs, adipocytes, myocytes, and fibroblasts, where they contribute to several cellular functions including transcytosis, cholesterol homeostasis, pathogen uptake, mechanotransduction, cell signaling, and cancer progression [171,172,173,174]. The increasing number of studies demonstrating the remarkable amount of caveolae in endothelium as well as the high expression of caveolins in ECs, along with the availability of caveolin-1-deficient animal models [175,176,177,178,179], have significantly contributed to shed light on the biological significance of the caveolar network in vascular functions [180,181,182,183,184,185,186,187,188,189]. A study by Razani and collaborators was among the first to prove the role of caveolin-1 in the regulation of lipid homeostasis and obesity in vivo [190]. In addition, the capacity of caveolin-1 to coat lipid droplets (i.e., specialized structures involved in intracellular lipid metabolism), as well as its role in the regulation of fatty acid metabolism and glycolysis have been demonstrated [191,192]. In cancer cells, a high expression of caveolin-1 influences the glycolytic rate by increasing glucose uptake, as well as ATP production by inducing glucose transporter 3 (GLUT3) transcription [193]. Although the association of glycolytic enzymes with cytoskeletal structures such as actin filaments and microtubules in red blood cells was reported [194,195,196], whether the caveolar network participates in the intracellular regulation and distribution of glycolytic enzymes remains unknown. However, the association between the glycolytic enzymes phosphofructokinase and aldolase with caveolin-1 that has been reported in smooth muscle cells strongly supports the existence of a membrane-bound glycolysis [197]. In addition, siRNA-mediated caveolin-1 knockdown has been shown to decrease the availability of glycolytic intermediates as well as to increase mitochondria-derived ROS in ECs [198], yet the contribution of the caveolar network to endothelial metabolic switch in response to changes occurring in the microenvironment still remains poorly understood. On the basis of these studies, it makes sense to hypothesize a role of caveolin-1 in regulating major metabolic pathways as well as the level of oxidative stress in ECs. We believe that improving our understanding of the complex metabolic processes involving the caveolar network in EC functions may open new avenues to develop novel therapeutic approaches targeting cardiovascular diseases.

## 10. Conclusions

Beyond its well-known histological classification, the endothelium shows morphological and functional differences based on anatomical and environmental context. The recent discoveries highlighting the role of metabolism in the regulation of the phenotypic profile of ECs have generated fresh enthusiasm in the field of vascular biology. To this regard, deepening the metabolomics of ECs from different anatomical contexts as well as from different physiological and pathological conditions would significantly contribute to revealing novel attractive targets aimed at counteracting endothelial dysfunction and inhibiting abnormal vessel growth in cancer. At least two main aspects deserve further investigation: (i) identifying the players that regulate the metabolic switch in ECs in response to incoming cues, and (ii) exploring how the metabolic status of ECs influences their interactions with the surrounding microenvironment.

## Figures and Tables

**Figure 1 cancers-14-01929-f001:**
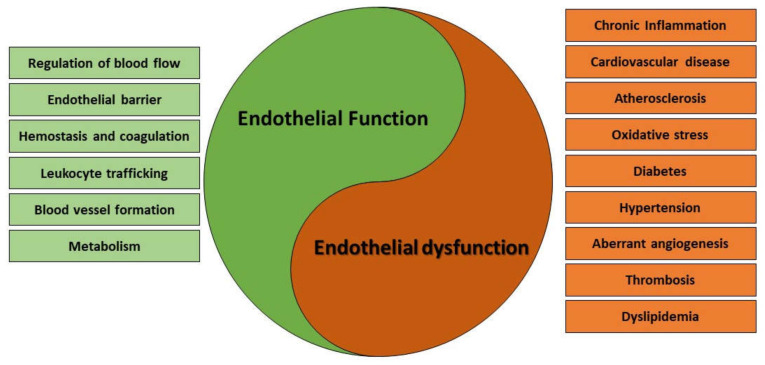
Main processes involved in endothelial functions and dysfucntions.

**Figure 2 cancers-14-01929-f002:**
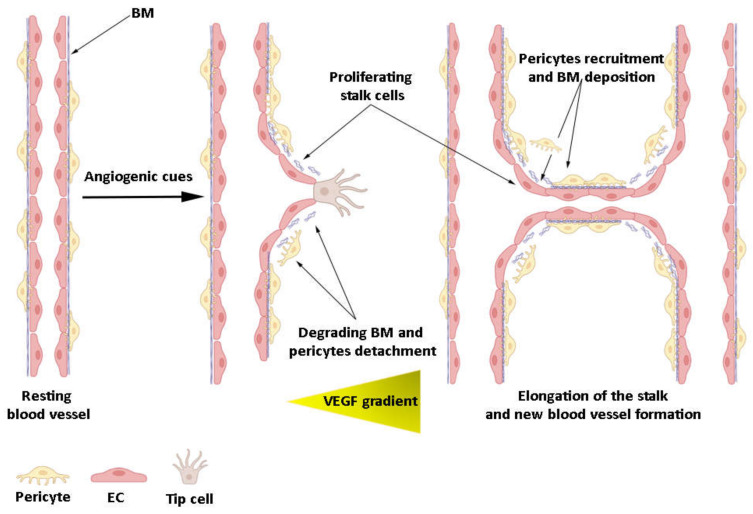
Main steps in vessel sprouting. In poorly perfused tissues ECs exposed to high VEGF concentration extend numerous filopodia and become tip cells, initiating sprouting angiogenesis. Degradation of BM and detachment of mural cells allow the stalk cells behind the tip cells to proliferate, contributing to elongating the nascent sprout. Finally, adjacent sprouts fuse and form the lumen. The new blood vessel is eventually stabilized by the recruitment of mural pericytes and deposition of BM (basement membrane).

**Figure 3 cancers-14-01929-f003:**
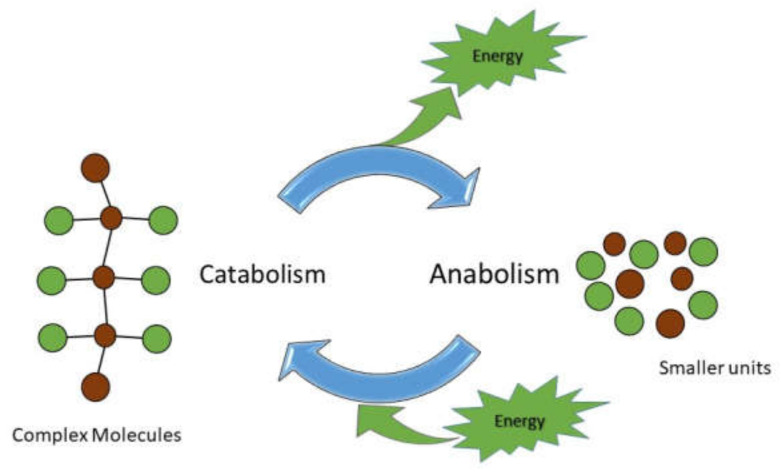
Metabolic pathways. Metabolism refers to the sum of all the chemical reactions in the body. Catabolism involves breaking down complex compounds into simpler ones, resulting in the release of chemical energy. Anabolism involves building larger and more complex chemical macromolecules from smaller subunits and requires energy obtained from ATP molecules.

**Table 1 cancers-14-01929-t001:** Food and Drug Administration (FDA)-approved angiogenesis inhibitors. Source, National Cancer Institute.

Name	Mechanism of Action and Target	Clinical Indications
Axitinib (Inlyta)	Inhibit receptor tyrosine kinases VEGFR-1, VEGFR-2, and VEGFR-3	Advanced renal cell carcinoma.
Bevacizumab (Avastin)	Prevents the interaction of VEGF to VEGFR1/Flt-1 and VEGFR2/KDR on the surface of ECs	Cervical cancer, Colorectal cancer, Glioblastoma, Hepatocellular carcinoma, Non-squamous non-small cell lung cancer, Renal cell carcinoma
Cabozantinib (Cometric)	Inhibits the tyrosine kinase activity of VEGFR-1, VEGFR-2, and other receptor tyrosine kinases.	Thyroid cancers, Hepatocellular carcinoma, Renal cell carcinoma
Everolimus (Afinitor)	Inhibits antigenic and interleukin (IL-2 and IL-15) stimulated activation and proliferation of T and B lymphocytes.	Astrocytoma, breast cancer, pancreatic cancer, gastrointestinal cancer, lung cancer, renal cell carcinoma
Pazopanib (Votrient)	Multi-tyrosine kinase inhibitor.	Renal Cell Carcinoma, Soft Tissue Sarcoma.
Ramucirumab (Cyramza)	VEGFR2 antagonist	Gastric Cancer, Non-Small Cell Lung Cancer, Colorectal Cancer, Hepatocellular Carcinoma
Regorafenib (Stivarga)	Surface and intracellular kinase inhibitor	Colorectal Cancer, Gastrointestinal Stromal Tumors, Hepatocellular Carcinoma
Sorafenib (Nexavar)	Surface and intracellular kinase inhibitor	Hepatocellular Carcinoma, Renal Cell Carcinoma, Thyroid Carcinoma
Sunitinib (Sutent)	Multiple receptor tyrosine kinases inhibitor (PDGFR, VEGFR1, VEGFR2, VEGFR3, etc)	Gastrointestinal Stromal Tumor, Advanced Renal Cell Carcinoma, Advanced Pancreatic Neuroendocrine Tumors
Vandetanib (Caprelsa)	Tyrosine kinase inhibitor	Medullary thyroid cancer
Ziv-Aflibercept (Zaltrap)	VEGF-A and VEGF-B inhibitor	Colorectal cancer

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
