# Peer review of "Endothelial Cell Metabolism in Vascular Functions"

_cancers, 2022, doi:10.3390/cancers14081929_

Round 1

Reviewer 1 Report

The Review manuscript by Filippini and colleagues provides a summary of relevant studies regarding metabolic plasticity and endothelial cell heterogeneity associated with various vascular functions.  The authors present an overview of the origin of endothelial cells; pertinent cellular metabolic pathways; endothelial cell activation; metabolic status in adult vasculature; angiogenesis and inflammation; viral infection; and involvement of plasma membrane microdomains.  Overall, this is a good Review.  To strengthen the presentation and impact of this study, the following suggestions are recommended:

  • For the readership of Cancers, a section dedicated to endothelial cells in cancer is highly recommended - - with a comment about the challenges and failure of anti-angiogenesis therapy.
  • An artistic rendition of endothelial cells involved in various vascular functions would be impactful as a summary figure.
  • The NOTCH family has a variety of types which should be mentioned.
  • There are many grammatical errors that should be addressed.

Author Response

For the readership of Cancers, a section dedicated to endothelial cells in cancer is highly recommended - with a comment about the challenges and failure of anti-angiogenesis therapy.

We thank the reviewer for her/his valuable suggestion. We have added a new section number 3 titled "Angiogenesis in cancer" (highlighted in yellow) to the revised version of the manuscript.

An artistic rendition of endothelial cells involved in various vascular functions would be impactful as a summary figure.

We agree with the reviewer that adding a schematic summary of the main functions of ECs contributes to enriching the article as well as to improve the overall readability of the manuscript.  Therefore we have added a new Figure 1 to specifically fulfill the reviewer's request. In addition, other cartoons and a table have been also included in the revised article.

The NOTCH family has a variety of types which should be mentioned.

We apologize with the reviewer for this evident slip-up carelessness. To this regard, a new sentence 88, and the relevant reference, have been introduced starting at line 285 of the revised manuscript concerning this topic.

There are many grammatical errors that should be addressed.

We would like to express our deep apologies again with the reviewer for our carelessness. We have revised the manuscript and corrected evident errors and typos. All changes made to the revised manuscript have been clearly indicated in yellow through the text.

Reviewer 2 Report

In a sense, this is an interesting and educational review that places recent advances in vascular biology in metabolomics in a historical context. This review is epic, more like a book than an academic review article. There are emotional expressions, many of which seem unnecessary or questionable for an academic paper.

Perhaps the authors intentionally did not include any figures and intended to convey details by letting the reader read the text without any preconceptions. However, that is not very kind to the reader, adds much cognition burden, and takes time to read. It is unfavorable as a modern paper to dare to put so much burden on the reader. Figures should be added to most sections.

Also, what does the title Metabolic plasticity refer to? I read to the end and did not find any mention of plasticity and irreversible metabolic changes. Why did the authors use the word plasticity in the title? The authors should describe what progress has been made in recent years in the study of Metabolic plasticity or Irreversibility and what has changed by the progress.

Is it cell-cell heterogeneity, intracellular heterogeneity or spatial heterogeneity? It is unclear whether the description assumes metabolomics in Bulk cells or Single-cell metabolomics.

There is a dissociation between what is expected from the title and the actual text.

The paper may be good in the sense that it outlines recent developments in metabolomics and communicates to non-specialists outside the field. However, the description of recent metabolomics is superficial and lacks a technical perspective. It makes me feel that the authors do not understand metabolomics data.

Author Response

In a sense, this is an interesting and educational review that places recent advances in vascular biology in metabolomics in a historical context. This review is epic, more like a book than an academic review article. There are emotional expressions, many of which seem unnecessary or questionable for an academic paper.

We thank the reviewer for her/his evident accuracy in reading and reviewing our article. We tried, at our best, to make the readability of the manuscript as smooth as possible. However, we hope it can be also suitable as well as easily readable for students and younger researchers who not fully familiar with these topics or are approaching the field of vascular biology.

Perhaps the authors intentionally did not include any figures and intended to convey details by letting the reader read the text without any preconceptions. However, that is not very kind to the reader, adds much cognition burden, and takes time to read. It is unfavorable as a modern paper to dare to put so much burden on the reader. Figures should be added to most sections.

We deeply apologize with the reviewer for not giving sufficient attention to the graphical details of our manuscript. We fully agree with her/him that introducing figures or illustrations in the original version of the manuscript could had contributed to an easier reading of the article. We have accordingly introduced three figures and one table in the revised version of the manuscript, hoping that these changes would fulfill the reviewer's requests.

Also, what does the title Metabolic plasticity refer to? I read to the end and did not find any mention of plasticity and irreversible metabolic changes. Why did the authors use the word plasticity in the title? The authors should describe what progress has been made in recent years in the study of Metabolic plasticity or Irreversibility and what has changed by the progress.

We agree with the reviewer that the original title was not perfectly exactly matched to the main topics. Therefore, we decided to change and simplify the title. We hope it would be now more fitting the topics described in the article.

Is it cell-cell heterogeneity, intracellular heterogeneity, or spatial heterogeneity? It is unclear whether the description assumes metabolomics in Bulk cells or Single-cell metabolomics.

We intended to point-out primarily the diversity of EC subtypes at the anatomical and morphological level. It is indeed well accepted that, behind the classical histological definition of endothelial tissue, ECs show specific differences in terms of morphology and responses to incoming stimuli (i.e. proinflammatory) that appear to be consistent to a specific gene expression pattern.

There is a dissociation between what is expected from the title and the actual text.

As previously mentioned, the title of the manuscript has been changed in the revised version of the manuscript.

The paper may be good in the sense that it outlines recent developments in metabolomics and communicates to non-specialists outside the field. However, the description of recent metabolomics is superficial and lacks a technical perspective. It makes me feel that the authors do not understand metabolomics data.

We would like to catch the reviewer's attention that it was not our aim to focus on metabolomics. To this regard, we have just mentioned this subject in section 10 "Conclusions" to point out the importance of this approach in the field of vascular endothelium, with the intent of leaving to the reader to deepen this topic.

Reviewer 3 Report

This paper, although well written, lacks structure and figures to allow a better understanding of the subject.
Moreover, taking into account the title of the article, a focus on the metabolic aspects in endothelial cells would be more judicious.
On the other hand, it would be necessary to take into account previous publications on the subject (Endothelial Cell Metabolism, G Eelen et al. 2018 Physiol Rev )  in order to highlight the new results obtained over the past four years.

Author Response

This paper, although well written, lacks structure and figures to allow a better understanding of the subject.

We apologize with the reviewer for not giving sufficient attention to the graphical details of our manuscript. We have accordingly introduced three figures and one table in the revised version of the manuscript.

Moreover, taking into account the title of the article, a focus on the metabolic aspects in endothelial cells would be more judicious.

We agree with the reviewer and accordingly changed the title as indicated in the revised version of the article.

On the other hand, it would be necessary to take into account previous publications on the subject (Endothelial Cell Metabolism, G Eelen et al. 2018 Physiol Rev )  in order to highlight the new results obtained over the past four years.

We are aware of the article suggested by the reviewer. To this regard, we would like to point out that it was already mentioned in the original submitted version of the manuscript (now at line 324 of the revised manuscript of Section 6).   

Round 2

Reviewer 1 Report

The authors have done an excellent job in revising this interesting review.  Please change "tha" to "that" in line 488.

Author Response

We thank the reviewer for her/his accuracy. We modified the text accordingly, as marked up using the "Track changes" function

Reviewer 2 Report

I am almost satisfied, except for the following points.

  1. The font size in the Figures is too small.
  2. References also should be included in Table 1.

Author Response

We agree with the reviewer that this changes will improve the readability of the article. We modified all the figures and the table according to the reviewer's suggestions.

Reviewer 3 Report

1- We thank the authors for the addition of figures and tables that facilitate the understanding of the article.

2-The modification of some paragraphs and the addition of the paragraph specific to cancer cells make the message of this review more clear and structured.

Author Response

We're glad to see that the revised version of the manuscript fully addressed the previous comments raised by the reviewer.